# Composition, determinants, and risk factors of low birth weight in Sri Lanka

**Sachith Mettananda**[1,2,3]*, **Himali Herath**[3,4], **Ayesha Thewage**[1],
**Kumudu Nanayakkara**[4], **Indeewari Liyanage**[4], **K. S. Udani**[4], **Rajika Savanadasa**[4],
**Sampatha Goonewardena**[3,5], **Nimesha Gamhewage**[3,6], **Asiri Hewamalage**[3,4],
**Dhammica Rowel**[7], **Abner Elkan Daniel**[7], **Chithramalee de Silva**[4], **Susie Perera**[3,8]

1 Department of Paediatrics, University of Kelaniya, Sri Lanka, 2 Colombo North Teaching Hospital, Sri
Lanka, 3 Perinatal Society of Sri Lanka, Sri Lanka, 4 Family Health Bureau, Sri Lanka, 5 Department of
Community Medicine, Faculty of Medical Sciences, University of Sri Jayewardenepura, Sri Lanka,
6 Department of Paediatrics, University of Sri Jayewardenepura, Sri Lanka, 7 UNICEF, Sri Lanka, 8 Ministry
of Health, Sri Lanka

* sachith.mettananda@kln.ac.lk

pone.0318554

& Gynecology Clinic Narodni Front, SERBIA

**Data Availability Statement:** All relevant data are
within the manuscript and its Supporting
information files.

## Abstract

### Introduction

Low birth weight continues to pose significant challenges to healthcare systems worldwide.
Despite substantial improvement in various public health indicators, many developing coun-
tries have failed to achieve a significant reduction in low birth weight rates. One major obsta-
cle is the sparsity of data on the determinants of low birth weight. Here, we aim to determine
the composition and risk factors for low birth weight in Sri Lanka, a prototype developing
nation.

### Methodology

We conducted a countrywide multicentre cross-sectional study in August and September
2023 in 13 purposively selected hospitals representing all nine provinces and different tiers
of specialist hospitals in Sri Lanka. All live-born neonates were recruited prospectively, and
their mothers were interviewed by trained data collectors to gather information on socio-
demographic background, medical and obstetric history, and delivery details. Birth weight
was measured immediately after the birth by trained healthcare personnel attending the
delivery.

### Results

A total of 9130 live-born neonates were recruited, of which 52% were males. The mean birth
weight was 2827g (95%CI 2817-2838g), and 1865 (20.4%) newborns were low birth weight.
The prevalence of prematurity was 10.9% (n = 998), and 1819 (20.0%) neonates were born
small for gestational age. Of the low birth weight neonates, 64% were small for gestational
age, and 37% were preterm; 11% were both small for gestational age and preterm. Teenage
pregnancy (p = 0.022), low maternal pre-pregnancy body mass index (p<0.001), inadequate
weight gain during pregnancy (p<0.001), maternal anaemia at delivery (p = 0.020), chronic

**Funding:** The study was funded by a grant awarded by UNICEF, Sri Lanka to the Perinatal Society of Sri Lanka and Family Health Bureau of Sri Lanka.

**Competing interests:** The authors have declared that no competing interests exist.

lung disease (p = 0.019), and pregnancy induced hypertension (p = 0.019) were significant modifiable risk factors for small for gestational age.

## Conclusion

This study presents the results of one of the most extensive country-wide studies evaluating the composition and determinants of low birth weight. The study highlights the importance of considering small for gestational age and prematurity as separate categories of low birth weight. Small for gestational age contributes to approximately two-thirds of the low birth weight burden. Therefore, targeting modifiable risk factors for small for gestational age while mitigating the burden of prematurity is the most feasible approach to reduce the prevalence of low birth weight in developing countries, including Sri Lanka.

## Introduction

Low birth weight (LBW), defined as birth weight less than 2500g, continues to pose significant challenges to healthcare systems worldwide [1]. It is estimated that nearly 20 million babies are born annually with LBW worldwide, resulting in a global prevalence of 15% [2]. The prevalence of LBW is one of the core health indicators of the World Health Organisation (WHO) as it reflects the long-term maternal nutritional status and health care during pregnancy. Reduction in the prevalence of LBW is an indicator of improvement in maternal health and nutritional status in a country. In 2012, the WHO and UNICEF jointly set a target to reduce the global prevalence of LBW by 30% between 2012 and 2025 [3]. However, many countries, especially low- and middle-income countries (LMICs), struggle to achieve this target and are expected to fall far behind the goal [4].

One major obstacle in achieving the target for LBW reduction is the lack of accurate data on the composition and risk factors for LBW in LMICs. The two main categories of LBW are prematurity and small for gestational age (SGA). Prematurity is defined as neonates born before 37 completed weeks of gestation, and the SGA is defined as having a birth weight below the 10th percentile for the gestational age based on sex-specific growth charts [5]. Preterm rupture of membranes, multiple pregnancy, and placental abruption are well recognised causes of prematurity, whereas pregnancy induced hypertension, chronic diabetes and maternal smoking are established causes of SGA [3, 6]. However, in many countries, the contribution of each cause to the prevalence of LBW is unknown. Also, many LMICs do not have the exact figures for LBW, mainly due to the high number of deliveries happening outside health facilities where birth weights are not accurately measured. Thus, the data on LBW from LMICs are primarily based on estimates rather than actual data [7].

Sri Lanka is a unique country in South Asia with a well-developed healthcare structure and improved health indicators compared to many LMICs [8]. Over 99% of deliveries occur in hospitals with trained medical and nursing staff, and the birth weights are measured accurately immediately after birth in every hospital [9, 10]. This contrasts with other South Asian countries like Bangladesh, Nepal and Pakistan, where only 40–70% of deliveries are reported to be institutional [11–13]. However, similar to other LMICs, the LBW prevalence has not decreased significantly over the last decade in Sri Lanka. The reasons behind this stagnant rate of LBW are understudied. Thus, Sri Lanka provides a valuable opportunity to understand the issues related to the high prevalence of LBW in LMICs. In this study, we aim to describe the

composition of LBW and to determine the risk factors for LBW in Sri Lanka. This should provide valuable insights into the determinants and risk factors of LBW in LMICs.

## Methodology

We conducted a countrywide multicentre cross-sectional study in Sri Lanka between 1 August to 30 September 2023. Thirteen hospitals representing all nine provinces and different tiers of specialist hospitals (i.e. Teaching, Provincial General, District General, and Base Hospitals) in Sri Lanka were purposively selected for the study to capture varied contexts within the country. The selection covered 20% of neonates born in the country and represented at least 15% of births in each province.

All live-born neonates born at the thirteen study sites during the study period were recruited after obtaining informed written consent from mothers. Stillbirths, conjoint twins and neonates transferred after birth from other hospitals were excluded. The sample size from each study centre was calculated based on the previously reported prevalence of LBW of 16% and for a 5% type 1 error. The minimum sample size required from each study site was 207.

Data were collected using an interviewer-administered questionnaire by interviewing mothers and perusing patient records. Trained medical-graduate research assistants interviewed mothers and collected data at thirteen study sites. The questionnaire contained questions on socio-demographic background, obstetric history of the index pregnancy, antenatal care received, past medical history, past obstetric history, delivery details and the immediate care of the newborn. Birth weight was measured immediately after the birth by trained healthcare personnel attending the delivery. The measurements were taken to the nearest 5g.

LBW was defined as weight at birth <2500g. Prematurity was defined as neonate delivered before completion of 37 weeks of gestation. The expected date of delivery determined based on the last regular menstrual period, and the dating ultrasound scan was considered to calculate the gestational age. Small for gestational age (SGA) was defined as birth weight below the 10th percentile for the sex and gestational age-specific international growth standards developed in the INTERGROWTH-21 study [14, 15].

Data were analysed using IBM SPSS Statistics 27.0. The prevalence of LBW and the relative contribution of prematurity and SGA for LBW were determined. Frequencies, percentages, means, and standard deviations were used to present descriptive statistics. The independent risk factors for prematurity and SGA were determined by logistic regression. Ethical clearance was obtained from the Ethics review committee of the Sri Lanka College of Paediatricians. Administrative approval was obtained from the Director General of Health Services.

## Results

A total of 9130 live-born neonates were recruited from 13 study sites over the two months (S1 Table in S1 File). A majority (52.3%) were males (S2 Table in S1 File).

### Sociodemographic, medical and pregnancy characteristics of the study population

A majority of the study population were Sinhalese (53.9%) and aged between 20–34 years (80.3%) (Table 1). Most neonates were delivered by vaginal delivery (55.5%); however, the caesarean section rate was 42.5%.

**Table 1. Sociodemographic, medical and pregnancy characteristics of the study population.**

| Characteristic | Frequency | Percentage |
|---|---|---|
| **SOCIODEMOGRAPHIC CHARACTERISTICS** | | |
| *Ethnicity* | *(N = 9130)* | |
| Sinhalese | 4924 | 53.9% |
| Sri Lankan Tamil | 1880 | 20.6% |
| Indian Tamil | 683 | 7.5% |
| Muslim | 1634 | 17.9% |
| Other | 9 | 0.1% |
| *Mother's age (years)* | *(N = 9126)* | |
| = <19 years (Teenage mothers) | 390 | 4.3% |
| 20–34 years | 7331 | 80.3% |
| = >35 years (Elderly mothers) | 1405 | 15.4% |
| *Marital status* | *(N = 9129)* | |
| Married | 9009 | 98.7% |
| Unmarried | 98 | 1.1% |
| Divorced / Separated | 14 | 0.2% |
| Widowed | 8 | 0.1% |
| *Mother's employment status* | *(N = 9111)* | |
| Housewife | 7154 | 78.4% |
| Working mother | 1957 | 21.5% |
| *Father's occupation* | *(N = 9049)* | |
| Unemployed | 40 | 0.4% |
| Unskilled worker | 1746 | 19.1% |
| Skilled worker | 3873 | 42.4% |
| Lower professional | 1690 | 18.5% |
| Higher professional | 181 | 2.0% |
| Business/self-employed | 1519 | 16.6% |
| *Mother's education level* | *(N = 9125)* | |
| No schooling | 22 | 0.2% |
| Only primary education | 665 | 7.3% |
| Up to Ordinary Level (Grade 11) | 4448 | 48.7% |
| Up to Advanced Level (Grade 13) | 2893 | 31.7% |
| Diploma or Degree | 1097 | 12.0% |
| *Father's education level* | *(N = 9070)* | |
| No schooling | 24 | 0.3% |
| Only primary education | 845 | 9.3% |
| Up to Ordinary Level (Grade 11) | 4853 | 53.2% |
| Up to Advanced Level (Grade 13) | 2621 | 28.7% |
| Diploma or Degree | 727 | 8.0% |
| *Monthly family income (LKR)* | *(N = 8869)* | |
| = <24999 | 446 | 4.9% |
| 25000–49999 | 3525 | 38.6% |
| 50000–99999 | 3501 | 38.3% |
| > = 100000 | 1397 | 15.3% |
| **MEDICAL CHARACTERISTICS** | | |
| *Pre-pregnancy medical conditions of the mother* | *(N = 9130)* | |
| Pre-gestational chronic diabetes | 213 | 2.3% |
| Chronic hypertension | 106 | 1.2% |
| Hypothyroidism | 296 | 3.2% |

*(Continued)*

**Table 1.** (Continued)

| Characteristic | Frequency | Percentage |
|---|:---:|:---:|
| Asthma | 640 | 7.0% |
| Other chronic lung diseases | 9 | 0.1% |
| Epilepsy | 80 | 0.9% |
| Psychiatric illness | 62 | 0.7% |
| Thalassaemia trait | 61 | 0.7% |
| Connective tissue diseases | 16 | 0.2% |
| Short stature (height <145cm) | 336/9110 | 3.7% |
| Underweight (BMI <18.5kg/m$^2$) | 1230/7976 | 15.4% |
| Obesity (BMI >30kg/m$^2$) | 729/7976 | 9.1% |
| Anaemia (booking visit haemoglobin <11.0g/dl) | 1966/8939 | 22.0% |
| **PREGNANCY CHARACTERISTICS** | | |
| *Parity* | *(N = 9126)* | |
| Primipara | 4188 | 45.9% |
| Multipara | 4938 | 54.1% |
| *Singleton or multiple pregnancy* | *(N = 9130)* | |
| Singleton | 8916 | 97.7% |
| Twin | 201 | 2.2% |
| Triplet | 13 | 0.1% |
| *Past obstetric history (in multipara)* | *(N = 4938)* | |
| Past history of prematurity | 268 | 5.4% |
| Past history of LBW | 886 | 17.9% |
| Last child being less than 2 years old | 255 | 5.1% |
| Previous neonatal deaths | 87 | 1.7% |
| Previous stillbirths | 80 | 1.6% |
| *Antenatal complications* | *(N = 9130)* | |
| Pregnancy-induced hypertension | 454 | 5.0% |
| Gestational diabetes | 890 | 9.7% |
| Urinary tract infection | 375 | 4.1% |
| Chorioamnionitis | 34 | 0.4% |
| Polyhydramnios | 94 | 1.0% |
| Oligohydramnios | 254 | 2.8% |
| Fetal growth restriction | 443 | 4.9% |
| Placenta previa | 74 | 0.8% |
| Placental abruption | 45 | 0.5% |
| Anaemia (haemoglobin <10.5g/dl at delivery) | 1734/9017 | 19.2% |
| *Mode of delivery* | *(N = 9130)* | |
| Normal vaginal delivery | 5064 | 55.5% |
| Forceps delivery | 90 | 1.0% |
| Vacuum delivery | 90 | 1.0% |
| Elective caesarean section | 1929 | 21.1% |
| Emergency caesarean section | 1957 | 21.4% |
| *Substance use during pregnancy (self-reported)* | *(N = 9130)* | |
| Smoking | 3 | 0.0% |
| Passive smoking | 689 | 7.5% |
| Alcohol use | 8 | 0.1% |
| Cannabis use | 2 | 0.0% |
| Heroin use | 4 | 0.0% |

## Distribution of birth weight

The mean birth weight of the study population was 2827g (95%CI 2817-2838g). The median birth weight was 2860g, and the birth weight range was 410g to 4820g. The mean birth weight of male neonates (2867±SD521g) was significantly higher than that of female neonates (2785 ±SD497g), (t = 7.6, p<0.001). The median birth weight varied between 2750g and 2950g in individual study sites (S3 Table in S1 File).

## Prevalence of LBW, prematurity and SGA

Overall, 1865 (20.4%) newborns had LBW. The prevalence of very LBW (birth weight between 1000-1499g) and extremely LBW (birth weight <1000g) were 1.1% and 0.8% respectively. The prevalence of prematurity was 10.9%, with 998 neonates being born preterm. The proportion of babies born moderate preterm (32–33 weeks), very preterm (28–31 weeks) and extreme preterm (<28 weeks) were 1.2%, 1.2% and 0.6% respectively. Of the extreme preterm neonates, one was delivered at 23 weeks. Number of neonates born at 24, 25, 26 and 27 weeks of gestation were 6 (0.1%), 16 (0.2%), 16 (0.2%) and 15 (0.2%), respectively (S1 Fig in S1 File).

We evaluated the adequacy of birth weight for respective gestational age using INTER-GROWTH-21 standards. We found that 1819 (20.0%) neonates were born SGA. The highest prevalence of SGA was reported at 41 weeks of gestation (43.8%), while the lowest prevalence (0%) was reported at 24 and 25 weeks of gestation (S4 Table in S1 File). The mode of delivery of LBW, preterm and SGA neonates is shown in S5 Table in S1 File.

## Composition of LBW

Out of 1860 LBW babies (excluding 5 LBW babies whose maturity or SGA could not be determined), 1182 (63.5%) were SGA, and 692 (37.2%) were premature; 214 (11.5%) were both SGA and premature (Fig 1). However, 200 (10.7%) LBW newborns were neither premature nor had SGA.

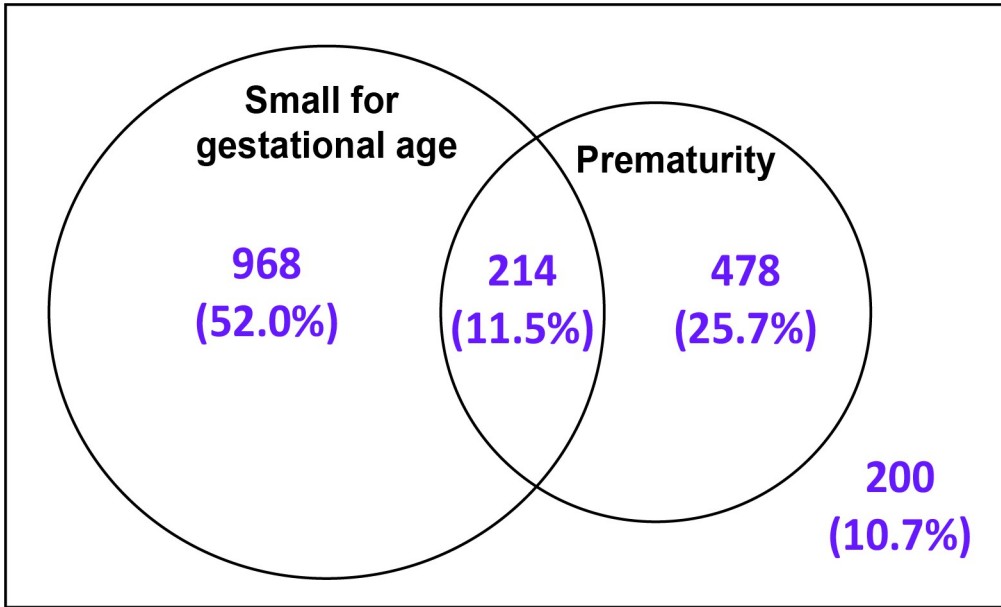

**Fig 1. Relative contribution of SGA and prematurity for LBW.**

Of the premature neonates, 304/996 (30.5%) had normal birth weight over 2500g. Similarly, of the SGA neonate, 637/1819 (35.0%) had a normal birth weight over 2500g (Fig 2).

### Risk factors for SGA

Next, we looked at the risk factors for SGA. Many socio-demographic and clinical characteristics of mothers were analysed by multivariate logistic regression to identify independent risk factors for SGA (Table 2). This identified the following as independent risk factors for SGA: multiple pregnancy, Indian Tamil ethnicity, teenage mother, lower maternal pre-pregnancy BMI ($<18.5$kg/m$^2$), maternal short stature ($<145$cm), inadequate weight gain during pregnancy ($<12$kg), maternal anaemia at delivery (haemoglobin $<10.5$g/dl), maternal chronic lung disease, pregnancy-induced hypertension, oligohydramnios, and past history of LBW.

### Risk factors for prematurity

Next, we looked at the risk factors for prematurity using multivariate logistic regression (Table 3). This identified the following as independent risk factors for prematurity: male

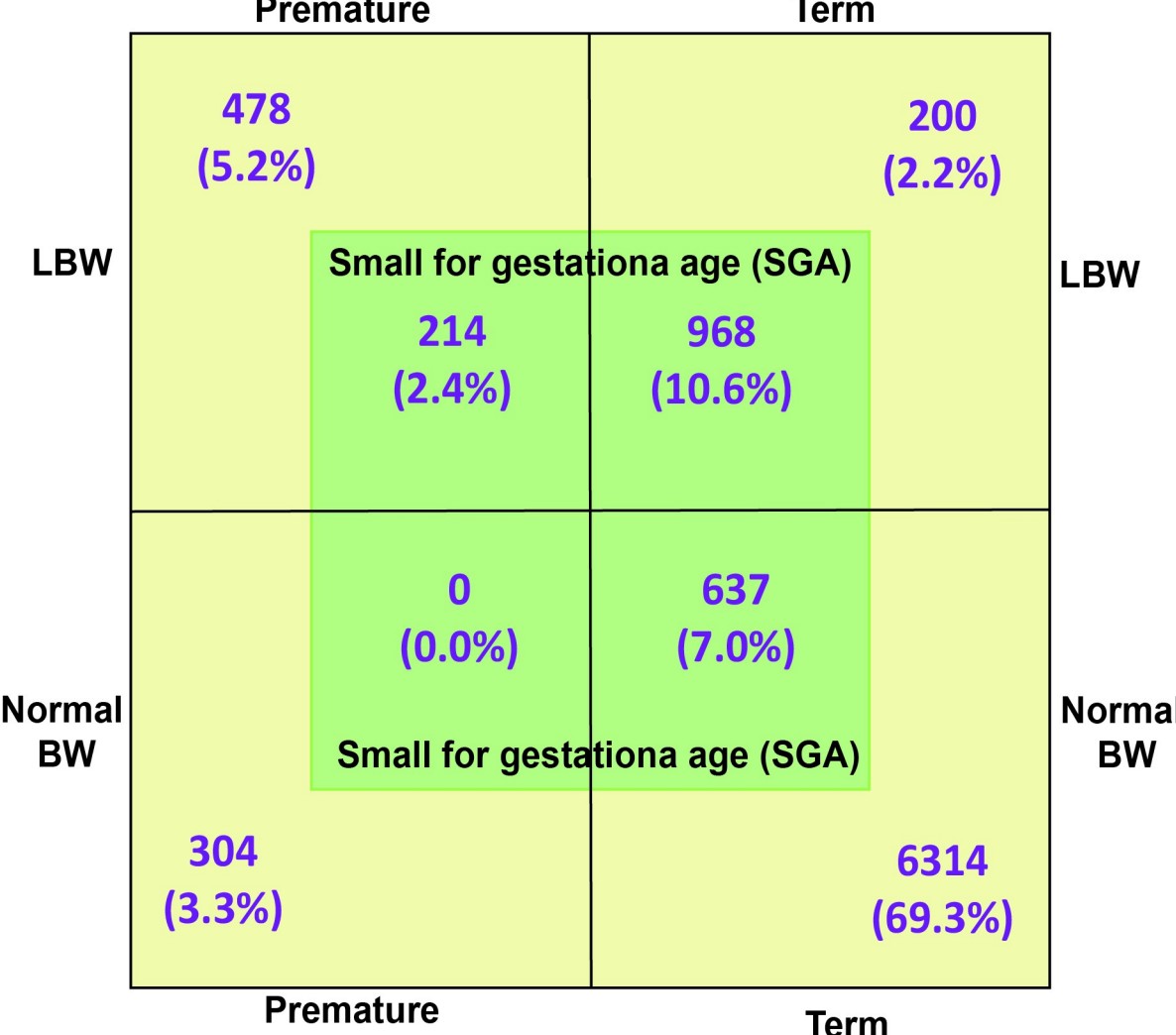

**Fig 2. Relative distribution of LBW, SGA and prematurity.**

**Table 2. Risk factors for small for gestational age.**

| Factor | Number (%) of neonates with SGA | Odds ratio (95%CI) [Unadjusted] | Adjusted odds ratio (95%CI)* | P value |
|---|---|---|---|---|
| Gender of the newborn | | | | |
| Female (N = 4342) | 896 (20.6%) | 1.08 (0.97–1.20) | 1.06 (0.93–1.20) | 0.334 |
| Male (N = 4767) | 922 (19.3%) | | | |
| Singleton or multiple pregnancy | | | | |
| Multiple pregnancy (N = 214) | 93 (43.5%) | 3.19 (2.42–4.20) | 4.55 (3.28–6.32) | <0.001 |
| Singleton pregnancy (N = 8901) | 1726 (19.4%) | | | |
| Ethnicity of mother | | | | |
| Indian Tamil ethnicity (N = 682) | 252 (37.0%) | 2.56 (2.17–3.02) | 1.82 (1.41–2.35) | <0.001 |
| Other ethnicities (N = 8433) | 1567 (18.6%) | | | |
| Marital status of parents | | | | |
| Unmarried (N = 98) | 23 (23.5%) | 1.23 (0.77–1.97) | 0.66 (0.31–1.40) | 0.282 |
| Married (N = 9019) | 1796 (19.9%) | | | |
| Teenage pregnancy | | | | |
| Teenage mother (N = 389) | 108 (27.8%) | 1.57 (1.25–1.97) | 1.44 (1.05–1.98) | 0.022 |
| Non-teenage mother (N = 8723) | 1711 (19.6%) | | | |
| Elderly pregnancy | | | | |
| Elderly mother (N = 1402) | 256 (18.3%) | 0.87 (0.75–1.01) | 0.83 (0.69–1.01) | 0.057 |
| Nonelderly mother (N = 7710) | 1563 (20.3%) | | | |
| Mother's employment status | | | | |
| Working mother (N = 1954) | 374 (19.1%) | 0.93 (0.82–1.06) | 1.12 (0.94–1.33) | 0.188 |
| Housewife (N = 7143) | 1443 (20.2%) | | | |
| Father's occupation | | | | |
| Non-professional (N = 7165) | 1446 (20.2%) | 1.09 (0.96–1.25) | 0.92 (0.77–1.09) | 0.360 |
| Professional (N = 1871) | 350 (18.7%) | | | |
| Mother's education level | | | | |
| OL or lower (N = 5125) | 1127 (22.0%) | 1.21 (1.07–1.36) | 1.07 (0.92–1.25) | 0.353 |
| AL or higher (N = 3987) | 692 (17.4%) | | | |
| Father's education level | | | | |
| OL or lower (N = 5712) | 1246 (21.8%) | 1.37 (1.23–1.53) | 1.15 (0.97–1.36) | 0.092 |
| AL or higher (N = 3345) | 565 (16.9%) | | | |
| Monthly family income | | | | |
| < LKR 50000 (N = 3964) | 883 (22.3%) | 1.30 (1.17–1.44) | 1.08 (0.94–1.25) | 0.252 |
| > = LKR 50000 (N = 4894) | 881 (18.0%) | | | |
| Maternal pre-pregnancy BMI | | | | |
| Underweight (N = 1228) | 343 (27.9%) | 1.87 (1.60–2.12) | 1.84 (1.56–2.16) | <0.001 |
| Not underweight (N = 6741) | 1169 (17.3%) | | | |
| Maternal height | | | | |
| <145cm (N = 335) | 115 (34.3%) | 2.18 (1.72–2.73) | 1.90 (1.41–2.54) | <0.001 |
| > = 145cm (N = 8767) | 1703 (19.4%) | | | |
| Weight gain in pregnancy | | | | |
| <12kg (N = 5010) | 1033 (20.6%) | 1.40 (1.22–1.60) | 1.50 (1.30–1.74) | <0.001 |
| > = 12kg (N = 2122) | 332 (15.6%) | | | |
| Maternal anaemia (at booking) | | | | |
| Yes (N = 1965) | 415 (21.1%) | 1.10 (0.97–1.25) | 0.94 (0.80–1.10) | 0.467 |
| No (N = 6967) | 1357 (19.5%) | | | |
| Maternal anaemia (at delivery) | | | | |
| Yes (N = 1728) | 320 (18.5%) | 0.89 (0.78–1.02) | 1.22 (1.03–1.45) | 0.020 |

*(Continued)*

**Table 2.** (Continued)

| Factor | Number (%) of neonates with SGA | Odds ratio (95%CI) [Unadjusted] | Adjusted odds ratio (95%CI)* | P value |
|---|---|---|---|---|
| No (N = 7274) | 1470 (20.2%) | | | |
| Maternal chronic diabetes | | | | |
| Yes (N = 212) | 23 (10.8%) | 0.48 (0.31–0.74) | 0.55 (0.33–0.91) | 0.022 |
| No (N = 8903) | 1796 (20.2%) | | | |
| Maternal chronic hypertension | | | | |
| Yes (N = 105) | 15 (14.3%) | 0.66 (0.38–1.15) | 0.94 (0.48–1.82) | 0.859 |
| No (N = 9007) | 1803 (20.0%) | | | |
| Maternal hypothyroidism | | | | |
| Yes (N = 295) | 53 (18.0%) | 0.87 (0.64–1.18) | 0.92 (0.63–1.35) | 0.688 |
| No (N = 8817) | 1765 (20.0%) | | | |
| Maternal asthma | | | | |
| Yes (N = 635) | 122 (19.2%) | 0.95 (0.77–1.16) | 1.12 (0.88–1.42) | 0.337 |
| No (N = 8477) | 1696 (20.0%) | | | |
| Maternal chronic lung disease | | | | |
| Yes (N = 9) | 4 (44.4%) | 3.21 (0.86–11.9) | 8.20 (1.42–47.2) | 0.019 |
| No (N = 9103) | 1814 (19.9%) | | | |
| Maternal epilepsy | | | | |
| Yes (N = 80) | 19 (23.8%) | 1.25 (0.74–2.10) | 1.13 (0.59–2.20) | 0.698 |
| No (N = 9032) | 1799 (19.9%) | | | |
| Maternal psychiatric illness | | | | |
| Yes (N = 62) | 14 (22.6%) | 1.17 (0.64–2.12) | 1.04 (0.49–2.23) | 0.907 |
| No (N = 9050) | 1804 (19.9%) | | | |
| Maternal thalassaemia trait | | | | |
| Yes (N = 61) | 8 (13.1%) | 0.60 (0.28–1.27) | 0.90 (0.41–1.99) | 0.804 |
| No (N = 9051) | 1810 (20.0%) | | | |
| Connective tissue disease | | | | |
| Yes (N = 15) | 2 (13.3%) | 0.61 (0.13–2.73) | 0.13 (0.01–1.41) | 0.095 |
| No (N = 9097) | 1816 (20.0%) | | | |
| Pregnancy-induced hypertension | | | | |
| Yes (N = 452) | 107(23.7%) | 1.25 (1.00–1.57) | 1.38 (1.05–1.81) | 0.019 |
| No (N = 8663) | 1712 (19.8%) | | | |
| Gestational diabetes | | | | |
| Yes (N = 889) | 127 (14.3%) | 0.64 (0.52–0.78) | 0.71 (0.56–0.90) | 0.005 |
| No (N = 8226) | 1692 (20.6%) | | | |
| Urinary tract infection | | | | |
| Yes (N = 375) | 68 (18.1%) | 0.88 (0.67–1.15) | 0.87 (0.63–1.20) | 0.403 |
| No (N = 8740) | 1751 (20.0%) | | | |
| Chorioamnionitis | | | | |
| Yes (N = 34) | 9 (26.5%) | 1.44 (0.67–3.10) | 1.63 (0.60–4.40) | 0.330 |
| No (N = 9081) | 1810 (19.9%) | | | |
| Polyhydramnios | | | | |
| Yes (N = 94) | 11 (11.7%) | 0.52 (0.28–0.99) | 0.41 (0.17–1.00) | 0.050 |
| No (N = 9021) | 1808 (20.0%) | | | |
| Oligohydramnios | | | | |
| Yes (N = 253) | 100 (39.5%) | 2.71 (2.09–3.51) | 2.67 (1.94–3.67) | <0.001 |
| No (N = 8862) | 1719 (19.4%) | | | |
| Placenta previa | | | | |

(*Continued*)

**Table 2.** (Continued)

| Factor | Number (%) of neonates with SGA | Odds ratio (95%CI) [Unadjusted] | Adjusted odds ratio (95%CI)* | P value |
|---|---|---|---|---|
| Yes (N = 74) | 8 (10.8%) | 0.48 (0.23–1.01) | 0.51 (0.19–1.32) | 0.168 |
| No (N = 9041) | 1811 (20.0%) | | | |
| Placental abruption | | | | |
| Yes (N = 45) | 7 (15.6%) | 0.73 (0.32–1.65) | 0.42 (0.14–1.28) | 0.131 |
| No (N = 9070) | 1812 (20.0%) | | | |
| Past history of prematurity | | | | |
| Yes (N = 267) | 64 (24.0%) | 1.27 (0.95–1.69) | 0.90 (0.61–1.31) | 0.596 |
| No (N = 8848) | 1755 (19.8%) | | | |
| Past history of LBW | | | | |
| Yes (N = 882) | 278 (31.5%) | 1.99 (1.71–2.32) | 1.92 (1.56–2.37) | <0.001 |
| No (N = 8233) | 1541 (18.7%) | | | |
| Age of youngest sibling <2 years | | | | |
| Yes (N = 255) | 37 (14.5%) | 0.67 (0.47–0.95) | 0.70 (0.44–1.10) | 0.128 |
| No (N = 8860) | 1782 (20.1%) | | | |
| Previous neonatal deaths | | | | |
| Yes (N = 87) | 21 (24.1%) | 1.27 (0.78–2.09) | 0.87 (0.48–1.59) | 0.671 |
| No (N = 9028) | 1768 (19.9%) | | | |
| Previous stillbirths | | | | |
| Yes (N = 80) | 18 (22.5%) | 1.16 (0.68–1.97) | 1.23 (0.65–2.32) | 0.508 |
| No (N = 9035) | 1801 (19.9%) | | | |
| Maternal passive smoking | | | | |
| Yes (N = 686) | 138 (20.1%) | 1.01 (0.83–1.22) | 0.93 (0.74–1.18) | 0.601 |
| No (N = 8429) | 1681 (19.9%) | | | |

* 6872 subjects with completed data were included in multivariate analysis.

gender, multiple pregnancy, elderly mother (> = 35 years), higher maternal education level (AL or above), lower maternal pre-pregnancy BMI (<18.5kg/m$^2$), maternal short stature (<145cm), inadequate weight gain during pregnancy (<12kg), chronic diabetes, chronic hypertension, pregnancy induced hypertension, oligohydramnios, placenta previa, placental abruption, past history of prematurity, and past history of stillbirths.

### Immediate neonatal outcome of LBW, premature and SGA neonates

Finally, we examined the immediate neonatal outcomes of participants in the study (Table 4). Of all newborns, 6.3% required resuscitation at birth, and the 5-minute APGAR score was <8 in 1.3%. Nine (0.1%) babies died soon after birth, and 6.3% were admitted to the neonatal intensive care unit (NICU). In 87% of neonates, breastfeeding was commenced within 1 hour of birth.

### Discussion

Despite advances in maternal care and public health, many countries have failed to achieve the desired reduction in LBW globally [4]. This is particularly true for LMICs, including Sri Lanka [16]. One significant obstacle in achieving this target is the sparsity of reliable large-scale data on the composition and determinants of LBW. A recent report highlights that the birth weights of 48% of newborns are not recorded globally [1]. The lack of data is primarily from

**Table 3. Risk factors for prematurity.**

| Factor | Number (%) of premature neonates | Odds ratio (95%CI) [Unadjusted] | Adjusted odds ratio (95%CI)* | P value |
|---|---|---|---|---|
| Gender of the newborn | | | | |
| Male (N = 4771) | 566 (11.9%) | 1.22 (1.07–1.39) | 1.36 (1.13–1.62) | <0.001 |
| Female (N = 4344) | 431 (9.9%) | | | |
| Singleton or multiple pregnancy | | | | |
| Multiple pregnancy (N = 214) | 143(66.8%) | 18.9 (14.1–25.4) | 23.3 (16.3–33.2) | <0.001 |
| Singleton pregnancy (N = 8903) | 855 (9.6%) | | | |
| Ethnicity | | | | |
| Indian Tamil ethnicity (N = 682) | 64 (9.4%) | 0.83 (0.63–1.08) | 1.08 (0.71–1.63) | 0.702 |
| Other ethnicities (N = 8440) | 934 (11.1%) | | | |
| Marital status of parents | | | | |
| Unmarried (N = 98) | 22 (22.4%) | 2.38 (1.47–3.85) | 1.30 (0.46–3.62) | 0.614 |
| Married (N = 9024) | 976 (10.8%) | | | |
| Teenage pregnancy | | | | |
| Teenage mother (N = 389) | 46 (11.8%) | 1.09 (0.80–1.50) | 1.16 (0.68–1.98) | 0.577 |
| Non-teenage mother (N = 8730) | 950 (10.9%) | | | |
| Elderly pregnancy | | | | |
| Elderly mother (N = 1404) | 222 (15.8%) | 1.68 (1.43–1.97) | 1.35 (1.07–1.69) | 0.009 |
| Nonelderly mother (N = 7715) | 774 (10.0%) | | | |
| Mother's employment status | | | | |
| Working mother (N = 1956) | 211 (10.8%) | 0.98 (0.83–1.15) | 1.02 (0.81–1.30 | 0.818 |
| Housewife (N = 7148) | 782 (10.9%) | | | |
| Father's occupation | | | | |
| Non-professional (N = 7172) | 786 (11.0%) | 1.04 (0.88–1.23) | 1.12 (0.87–1.43) | 0.357 |
| Professional (N = 1871) | 197 (10.5%) | | | |
| Mother's education level | | | | |
| AL or higher (N = 3989) | 455 (11.4%) | 1.09 (0.95–1.24) | 1.33 (1.07–1.65) | 0.009 |
| OL or lower (N = 5130) | 541 (10.5%) | | | |
| Father's education level | | | | |
| AL or higher (N = 3347) | 364 (10.9%) | 0.99 (0.87–1.14) | 0.90 (0.72–1.14) | 0.413 |
| OL or lower (N = 5717) | 623 (10.9%) | | | |
| Monthly family income | | | | |
| < LKR 50000 (N = 3969) | 428 (10.8%) | 1.10 (0.87–1.14) | 1.07 (0.87–1.31) | 0.515 |
| > = LKR 50000 (N = 4896) | 528 (10.8%) | | | |
| Maternal pre-pregnancy BMI | | | | |
| Underweight (N = 1230) | 138 (11.2%) | 1.09 (0.89–1.32) | 1.45 (1.13–1.85) | 0.003 |
| Not underweight (N = 6746) | 701 (10.4%) | | | |
| Maternal height | | | | |
| <145cm (N = 336) | 49 (14.6%) | 1.41 (1.03–1.93) | 1.57 (1.05–2.37) | 0.028 |
| > = 145cm (N = 8773) | 943 (10.7%) | | | |
| Weight gain in pregnancy | | | | |
| <12kg (N = 5015) | 537 (10.7%) | 1.46 (1.21–1.75) | 1.54 (1.25–1.91) | <0.001 |
| >12kg (N = 2123) | 161 (7.6%) | | | |
| Maternal anaemia (at booking) | | | | |
| Yes (N = 1966) | 211(10.7%) | 0.99 (0.84–1.16) | 0.98 (0.78–1.23) | 0.896 |
| No (N = 6973) | 752 (10.8%) | | | |
| Maternal anaemia (at delivery) | | | | |
| Yes (N = 1730) | 203 (11.7%) | 1.12 (0.95–1.32) | 1.00 (0.79–1.26) | 0.976 |

(*Continued*)

**Table 3.** (*Continued*)

| Factor | Number (%) of premature neonates | Odds ratio (95%CI) [Unadjusted] | Adjusted odds ratio (95%CI)* | P value |
|---|---|---|---|---|
| No (N = 7279) | 771 (10.6%) | | | |
| Maternal chronic diabetes | | | | |
| Yes (N = 212) | 59 (27.8%) | 3.27 (2.40–4.45) | 2.53 (1.65–3.88) | <0.001 |
| No (N = 8910) | 939 (10.5%) | | | |
| Maternal chronic hypertension | | | | |
| Yes (N = 105) | 33 (31.4%) | 3.83 (2.52–5.81) | 4.30 (2.46–7.51) | <0.001 |
| No (N = 9014) | 963 (10.7%) | | | |
| Maternal hypothyroidism | | | | |
| Yes (N = 296) | 41 (13.9%) | 1.32 (0.94–1.85) | 0.94 (0.58–1.50) | 0.796 |
| No (N = 8823) | 955 (10.8%) | | | |
| Maternal asthma | | | | |
| Yes (N = 638) | 68 (10.7%) | 0.97 (0.74–1.26) | 0.85 (0.60–1.20) | 0.363 |
| No (N = 8481) | 928 (10.9%) | | | |
| Maternal chronic lung disease | | | | |
| Yes (N = 9) | 1 (11.1%) | 1.01 (0.12–8.16) | 0.81 (0.05–12.6) | 0.882 |
| No (N = 9110) | 995 (10.9%) | | | |
| Maternal epilepsy | | | | |
| Yes (N = 80) | 9 (11.3%) | 1.03 (0.51–2.07) | 0.92 (0.33–2.58) | 0.888 |
| No (N = 9039) | 987 (10.9%) | | | |
| Maternal psychiatric illness | | | | |
| Yes (N = 62) | 6 (9.7%) | 0.87 (0.37–2.03) | 0.83 (0.28–2.47) | 0.741 |
| No (N = 9057) | 990 (10.9%) | | | |
| Maternal thalassaemia trait | | | | |
| Yes (N = 61) | 7 (11.5%) | 1.05 (0.48–2.33) | 1.23 (0.44–3.39) | 0.684 |
| No (N = 9058) | 989 (10.9%) | | | |
| Connective tissue disease | | | | |
| Yes (N = 16) | 2 (12.5%) | 1.16(0.26–5.13) | 0.55 (0.08–3.71) | 0.547 |
| No (N = 9103) | 994 (10.9%) | | | |
| Pregnancy-induced hypertension | | | | |
| Yes (N = 454) | 148 (32.6%) | 4.44 (3.61–5.48) | 4.92 (3.75–6.44) | <0.001 |
| No (N = 8668) | 850 (9.8%) | | | |
| Gestational diabetes | | | | |
| Yes (N = 890) | 123 (13.8%) | 1.34 (1.10–1.65) | 1.26 (0.96–1.65) | 0.088 |
| No (N = 8232) | 875 (10.6%) | | | |
| Urinary tract infection | | | | |
| Yes (N = 375) | 55 (14.7%) | 1.42 (1.06–1.90) | 1.31 (0.88–1.95) | 0.170 |
| No (N = 8747) | 943 (10.8%) | | | |
| Chorioamnionitis | | | | |
| Yes (N = 34) | 13 (38.2%) | 5.09 (2.54–10.20) | 2.86 (0.94–8.62) | 0.062 |
| No (N = 9088) | 985 (10.8%) | | | |
| Polyhydramnios | | | | |
| Yes (N = 94) | 19 (20.2%) | 2.08 (1.25–3.46) | 1.42 (0.70–2.87) | 0.328 |
| No (N = 9028) | 979 (10.8%) | | | |
| Oligohydramnios | | | | |
| Yes (N = 253) | 45 (17.8%) | 1.79 (1.29–2.49) | 2.05 (1.32–3.18) | 0.001 |
| No (N = 8869) | 953 (10.7%) | | | |
| Placenta previa | | | | |

(*Continued*)

**Table 3.** (Continued)

| Factor | Number (%) of premature neonates | Odds ratio (95%CI) [Unadjusted] | Adjusted odds ratio (95%CI)* | P value |
|---|---|---|---|---|
| Yes (N = 74) | 33 (44.6%) | 6.74 (4.24–10.71) | 6.68 (3.58–12.4) | <0.001 |
| No (N = 9048) | 965 (10.7%) | | | |
| Placental abruption | | | | |
| Yes (N = 45) | 28 (62.2%) | 13.7 (7.5–25.2) | 5.82 (2.41–14.0) | <0.001 |
| No (N = 9077) | 970 (10.7%) | | | |
| Past history of prematurity | | | | |
| Yes (N = 268) | 74 (27.6%) | 3.27 (2.48–4.31) | 2.84 (1.87–4.32) | <0.001 |
| No (N = 8854) | 924 (10.4%) | | | |
| Past history of LBW | | | | |
| Yes (N = 885) | 149 (16.8%) | 1.76 (1.45–2.13) | 1.14 (0.84–1.55) | 0.387 |
| No (N = 8237) | 849 (10.3%) | | | |
| Age of youngest sibling <2 years | | | | |
| Yes (N = 255) | 30 (11.8%) | 1.08 (0.73–1.60) | 1.05 (0.61–1.81) | 0.845 |
| No (N = 8867) | 968 (10.9%) | | | |
| Previous neonatal deaths | | | | |
| Yes (N = 87) | 20 (23.0%) | 2.45 (1.48–4.07) | 1.35 (0.69–2.65) | 0.371 |
| No (N = 9035) | 978 (10.8%) | | | |
| Previous stillbirths | | | | |
| Yes (N = 80) | 20 (25.0%) | 2.74 (1.65–4.57) | 3.07 (1.59–5.93) | <0.001 |
| No (N = 9042) | 978 (10.8%) | | | |
| Maternal passive smoking | | | | |
| Yes (N = 686) | 75 (10.9%) | 0.99 (0.77–1.28) | 1.08 (0.78–1.51) | 0.617 |
| No (N = 8436) | 923 (10.9%) | | | |

* 6877 subjects with all completed data were included in the analysis

LMIC, where babies are born outside health care institutes or in hospitals without adequate facilities and due to weak health information systems. Here, we presented the results of one of the most comprehensive and country-wide studies evaluating the causes, composition, and risk factors of LBW in a LMIC.

**Table 4. Immediate neonatal outcome of LBW, premature and SGA neonates.**

| Characteristic | Frequency (%) | | | |
|---|---|---|---|---|
| | All neonates | LBW group | Preterm group | SGA group |
| Resuscitated at birth | 574/9130 (6.3%) | 273/1865 (14.6%) *** | 235/998 (23.5%) *** | 139/1819 (7.6%) * |
| 5-minute APGAR <8 | 116/9125 (1.3%) | 63/1862 (3.4%) *** | 55/996 (5.5%) *** | 29/1817 (1.6%) |
| Admitted to the NICU | 571/9130 (6.3%) | 385/1865 (20.6%) *** | 365/998 (36.6%) *** | 152/1819 (8.4%) ** |
| Breastfeeding not initiated within 1 hour after birth | 1162/9129 (12.7%) | 515/1865 (27.6%) *** | 431/998 (43.1%) *** | 289/1819 (15.9%) *** |

* p<0.05,
** p<0.01,
*** p<0.001; compared to all neonates

This study was conducted through the prospective collection of data from 13 hospitals across Sri Lanka covering all nine provinces and different tiers of hospitals in the country. This contrasts with the previously reported studies, either done by secondary data analysis or performed in small and selected areas [17–20]. As over 99% of births in Sri Lanka are institutional deliveries happening in hospitals, the data in our study represent a unique cohort of newborns where accurate measurement of birth weights and assessment of gestational ages are carried out in a single LMIC. The study sites represented over 20% of births in the country during the study period.

Our results show that the overall prevalence of LBW in the study sites is 20.4%. Although the purposive selection of the study sites could represent a slight overrepresentation of LBW births, our results confirm that the LBW prevalence in Sri Lanka has not declined. This has been observed in many other LMICs, which show stagnation of LBW rates [21]. This could be due to many reasons, including the impact of the COVID-19 pandemic, recent economic depression and other medical and pregnancy-related factors. Hence, the reliable identification of the composition, determinants, and risk factors of LBW described in this paper is critical to develop strategies to reduce LBW in LMICs.

We found that the LBW in Sri Lanka is contributed mainly by SGA rather than prematurity. Our results show that 64% of LBW were SGA, while only 37% were premature. This contrasts with developed and more affluent countries where prematurity contributes to most LBW [22, 23]. Our results accurately represent the situation in developing LMICs, which contribute to the largest proportion of LBW globally. Therefore, LMICs with high LBW rates should concentrate on reducing the burden of SGA while mitigating the burden of prematurity.

Another important strength of our study is that we identified the risk factors for SGA and prematurity separately. Although risk factors for LBW and prematurity are well established, data on independent risk factors for SGA as a separate group determined through large-scale country-wide prospective studies in developing countries are lacking [24]. We found several non-modifiable (multiple pregnancy, Indian Tamil ethnicity, maternal short stature, past history of LBW) and modifiable (teenage pregnancy, low maternal pre-pregnancy BMI, inadequate weight gain during pregnancy, maternal anaemia at delivery, chronic lung disease, pregnancy induced hypertension) risk factors as significant independent predictors of SGA. These modifiable risk factors can be targeted to implement control programs. Minimising teenage pregnancies by discouraging underage marriages, improving the pre-pregnancy BMI of underweight women by nutritional intervention for women of childbearing age, improving weight gain during pregnancy by stringent monitoring and dietary interventions, avoiding maternal anaemia at delivery by iron and folate or multiple micronutrient supplementation, improving maternal and paternal education levels and increasing family income by economic empowerment are recommended strategies to reduce the burden of LBW due to SGA.

The independent risk factors for prematurity identified in the study are male gender, multiple pregnancy, elderly mother, higher maternal education level, lower maternal pre-pregnancy BMI, maternal short stature, inadequate weight gain during pregnancy, chronic diabetes, chronic hypertension, pregnancy-induced hypertension, oligohydramnios, placenta previa, placental abruption, past history of prematurity, and past history of stillbirths. These are consistent with the previously recognised risk factors in developed and developing countries. Thus, minimising elderly pregnancies and pregnancies in unmarried women, improving pre-pregnancy BMI of underweight women, monitoring and improving weight gain during pregnancy, optimal control of chronic diabetes and chronic hypertension, and treatment of urinary tract infection and chorioamnionitis are recommended strategies to reduce the burden of LBW due to prematurity.

Another observation of the study was that over 10% of neonates with LBW (birth weight <2500g) were neither SGA nor premature. This is because the 10th percentile for SGA at 37 completed weeks (considered term) is below the 2500g cut-off for LBW. The SGA cut-offs for boys and girls at 37 weeks were 2380g and 2330g, respectively. This creates confusion among the health care providers and is a concern. It highlights the need for reconsideration of the cut-off weights to diagnose LBW. It may be appropriate to have separate sex-specific cut-off values for LBW, which are not higher than the cut-off for SGA at 37 completed weeks.

Finally, we looked at the immediate neonatal outcomes of LBW, premature and SGA babies. It showed that a significantly higher proportion of LBW, premature, and SGA neonates required neonatal intensive care. In addition to having a higher incidence of neonatal complications, babies who were LBW due to SGA are known to have a higher prevalence of chronic non-communicable diseases, including metabolic syndrome, in adult life [25, 26]. This association is less pronounced in neonates who develop LBW due to prematurity. This further highlights the importance of considering prematurity and SGA as separate categories rather than considering LBW as a single group when making policy decisions related to LBW.

One limitation of our study is the relatively higher elective caesarean section rate compared to developed countries [27]. Although early delivery by elective caesarean section is associated with LBW, it was not considered a risk factor. As elective caesarean sections are performed for already determined fetal growth restriction in utero and other indications requiring preterm delivery, elective caesarean section was not included in the regression models evaluating for independent risk factors of SGA and prematurity.

The WHO and UNICEF have set a goal to reduce the prevalence of LBW by 30% between 2012 and 2025. Our data indicates that SGA contributes to approximately two-thirds (64%) of LBW prevalence in Sri Lanka, while prematurity contributes only to one-third (37%). Thus, interventions that reduce 50% of preterm births will reduce the LBW rate by only 18–19%, whereas interventions that decrease the prevalence of SGA by 50% will reduce the LBW rate by 32%. Thus, it is more realistic and cost-effective to target SGA risk factors than prematurity risk factors to reduce the LBW prevalence in Sri Lanka.

In conclusion, we have presented the results of one of the most extensive country-wide studies evaluating the composition and determinants of LBW in Sri Lanka. The study highlights the importance of considering SGA and prematurity as separate groups causing LBW. The contribution of SGA for LBW is twice that of prematurity. Therefore, LMICs, including Sri Lanka, should target the modifiable risk factors for SGA while mitigating the burden of prematurity if these countries were to meet the UNICEF goal of reducing the prevalence of LBW.

## Supporting information

**S1 File. Supplemental figures and tables.**
(PDF)

**S2 File. Data set.**
(XLSX)

## Acknowledgments

We thank the staff of the Perinatal Society of Sri Lanka and the Family Health Bureau of Sri Lanka for providing administrative support for the project.

## Author Contributions

**Conceptualization:** Sachith Mettananda, Himali Herath, Nimesha Gamhewage, Asiri Hewamalage, Susie Perera.

**Data curation:** Sachith Mettananda, Himali Herath, Ayesha Thewage, Kumudu Nanayakkara, Indeewari Liyanage, K. S. Udani, Rajika Savanadasa.

**Formal analysis:** Sachith Mettananda, Ayesha Thewage.

**Funding acquisition:** Sachith Mettananda, Himali Herath, Asiri Hewamalage, Chithramalee de Silva, Susie Perera.

**Methodology:** Sachith Mettananda, Himali Herath, Sampatha Goonewardena, Asiri Hewamalage, Dhammica Rowel, Abner Elkan Daniel, Chithramalee de Silva, Susie Perera.

**Project administration:** Sachith Mettananda, Himali Herath, Ayesha Thewage, Kumudu Nanayakkara, Indeewari Liyanage, K. S. Udani, Rajika Savanadasa, Chithramalee de Silva, Susie Perera.

**Resources:** Sachith Mettananda, Dhammica Rowel, Abner Elkan Daniel.

**Supervision:** Sachith Mettananda, Himali Herath, Sampatha Goonewardena, Chithramalee de Silva, Susie Perera.

**Writing – original draft:** Sachith Mettananda, Himali Herath, Chithramalee de Silva, Susie Perera.

**Writing – review & editing:** Sachith Mettananda, Himali Herath, Ayesha Thewage, Kumudu Nanayakkara, Indeewari Liyanage, K. S. Udani, Rajika Savanadasa, Sampatha Goonewardena, Asiri Hewamalage, Dhammica Rowel, Abner Elkan Daniel, Chithramalee de Silva, Susie Perera.

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
