## [Decision Letter · Decision Letter 0]

25 Oct 2024

PONE-D-24-07451Composition, determinants, and risk factors of low birth weight in Sri LankaPLOS ONE

Dear Dr. Mettananda,

Thank you for submitting your manuscript to PLOS ONE. After careful consideration, we feel that it has merit but does not fully meet PLOS ONE’s publication criteria as it currently stands. Therefore, we invite you to submit a revised version of the manuscript that addresses the points raised during the review process.

We look forward to receiving your revised manuscript.

Kind regards,

Muttaquina Hossain, MBBS, MPH

Academic Editor

PLOS ONE

“The study was funded by a grant awarded by UNICEF, Sri Lanka to the Perinatal Society of Sri Lanka and Family Health Bureau of Sri Lanka.”

“We thank the staff of the Perinatal Society of Sri Lanka and the Family Health Bureau of Sri Lanka for providing administrative support for the project. The study was funded by UNICEF, Sri Lanka.”

“The study was funded by a grant awarded by UNICEF, Sri Lanka to the Perinatal Society of Sri Lanka and Family Health Bureau of Sri Lanka.”

4. Please remove your figures from within your manuscript file, leaving only the individual TIFF/EPS image files, uploaded separately. These will be automatically included in the reviewers’ PDF.

5. We note that Supplemental Figure 1 in your submission contain [map/satellite] images which may be copyrighted. All PLOS content is published under the Creative Commons Attribution License (CC BY 4.0), which means that the manuscript, images, and Supporting Information files will be freely available online, and any third party is permitted to access, download, copy, distribute, and use these materials in any way, even commercially, with proper attribution. For these reasons, we cannot publish previously copyrighted maps or satellite images created using proprietary data, such as Google software (Google Maps, Street View, and Earth). For more information, see our copyright guidelines: http://journals.plos.org/plosone/s/licenses-and-copyright.

a. You may seek permission from the original copyright holder of Supplemental Figure 1 to publish the content specifically under the CC BY 4.0 license. 

Reviewers' comments:

Reviewer's Responses to Questions

**Comments to the Author**

1. Is the manuscript technically sound, and do the data support the conclusions?

Reviewer #1: Yes

Reviewer #2: Yes

2. Has the statistical analysis been performed appropriately and rigorously? 

Reviewer #1: Yes

Reviewer #2: Yes

3. Have the authors made all data underlying the findings in their manuscript fully available?

Reviewer #1: Yes

Reviewer #2: Yes

4. Is the manuscript presented in an intelligible fashion and written in standard English?

Reviewer #1: Yes

Reviewer #2: Yes

5. Review Comments to the Author

Reviewer #1: This study provides important findings in the field of obstetrics and neonatal health.

accept for publication

Revise and proofread the abstract, along with incorporating the results of P-values to strengthen the presentation.

Reviewer #2: The cross sectional study is a valuable guide to the factors that contribute to low birth weight in Sri Lanka. There are some minor clarifications that should improve the applicability of their findings.

1. The authors refer to “small for gestational age” as a “cause” of low birth weight in the Introduction, but it is really a category, not a cause. The authors could begin by briefly describing the likely predominant causes of babies that are born SGA, which they analyze in detail during the Results and Discussion.

2. Authors’ characterize the dataset from Sri Lanka as being unique among South Asian countries because of the high rate of births in health care facilities, but they cite no direct measurement from other South Asian countries.

3. There is no consideration of the high “elective” Cesarean-section rate among study subjects on birth weight, which has been shown in some studies to increase LBW.

4. Authors found that Tamil ethnicity was an independent predictor for LBW. Were they able to determine whether nutritional status of the mothers was the major driver? Given the disparity in socioeconomic status between Tamil and Sinhalese Sri Lankans, this seems to be an obvious consideration.

6. PLOS authors have the option to publish the peer review history of their article (what does this mean?). If published, this will include your full peer review and any attached files.

Reviewer #1: No

Reviewer #2: No

---

## [Author Response · Author response to Decision Letter 0]

28 Oct 2024

Responses to Reviewer 1 Comments:

1. This study provides important findings in the field of obstetrics and neonatal health. Accept for publication.

Author response: Thank you very much for this positive comment. 

2. Revise and proofread the abstract, along with incorporating the results of P-values to strengthen the presentation.

Author response: We have revised the abstract by proofreading and incorporating the results of P-values as suggested. 

Responses to Reviewer 2 Comments:

1. The cross sectional study is a valuable guide to the factors that contribute to low birth weight in Sri Lanka. 

Author response: Thank you very much for this positive comment. 

2. The authors refer to “small for gestational age” as a “cause” of low birth weight in the Introduction, but it is really a category, not a cause. The authors could begin by briefly describing the likely predominant causes of babies that are born SGA, which they analyze in detail during the Results and Discussion.

Author response: Thank you very much for the suggestion. We have revised the introduction as suggested. 

3. Authors’ characterize the dataset from Sri Lanka as being unique among South Asian countries because of the high rate of births in health care facilities, but they cite no direct measurement from other South Asian countries.

Author response: We have revised the introduction by including institutional delivery rates of other South Asian countries. 

4. There is no consideration of the high “elective” Cesarean-section rate among study subjects on birth weight, which has been shown in some studies to increase LBW.

Author response: We have included a new Supplementary Table (Supplementary Table 5) showing mode of delivery of LBW, Preterm and SGA neonates. 

5. Authors found that Tamil ethnicity was an independent predictor for LBW. Were they able to determine whether nutritional status of the mothers was the major driver? Given the disparity in socioeconomic status between Tamil and Sinhalese Sri Lankans, this seems to be an obvious consideration.

Author response: We included the variables like maternal pre-pregnancy BMI and weight gain in pregnancy that evaluate the nutrition status in the regression analysis. Indian Tamil ethnicity, maternal pre-pregnancy BMI, and maternal weight gain in pregnancy were all independent predictors of SGA indicating that the Indian Tamil ethnicity is a predictor independent of nutritional status.

---

## [Decision Letter · Decision Letter 1]

30 Dec 2024

PONE-D-24-07451R1Composition, determinants, and risk factors of low birth weight in Sri LankaPLOS ONE

Dear Dr. Mettananda,

Thank you for submitting your manuscript to PLOS ONE. After careful consideration, we feel that it has merit but does not fully meet PLOS ONE’s publication criteria as it currently stands. Therefore, we invite you to submit a revised version of the manuscript that addresses the points raised during the review process.

We look forward to receiving your revised manuscript.

Kind regards,

Tamara Sljivancanin Jakovljevic

Academic Editor

PLOS ONE

**Journal Requirements:**

Reviewers' comments:

Reviewer's Responses to Questions

**Comments to the Author**

1. If the authors have adequately addressed your comments raised in a previous round of review and you feel that this manuscript is now acceptable for publication, you may indicate that here to bypass the “Comments to the Author” section, enter your conflict of interest statement in the “Confidential to Editor” section, and submit your "Accept" recommendation.

Reviewer #2: (No Response)

Reviewer #3: (No Response)

Reviewer #4: All comments have been addressed

2. Is the manuscript technically sound, and do the data support the conclusions?

Reviewer #2: Yes

Reviewer #3: Yes

Reviewer #4: Yes

3. Has the statistical analysis been performed appropriately and rigorously? 

Reviewer #2: Yes

Reviewer #3: Yes

Reviewer #4: Yes

4. Have the authors made all data underlying the findings in their manuscript fully available?

Reviewer #2: Yes

Reviewer #3: Yes

Reviewer #4: Yes

5. Is the manuscript presented in an intelligible fashion and written in standard English?

Reviewer #2: Yes

Reviewer #3: Yes

Reviewer #4: Yes

6. Review Comments to the Author

**Reviewer #2:** Since the elective c-section rate reported is 20% nearly double the rate reported in high-resource countries, and delivery by c-section is associated with LBW, the authors should analyze whether or not elective c-section was an independent risk factor, or at a minimum acknowledge that this is a limitation.

**Reviewer #3: **The bottom line for me is that this study has not added any information that we do not previously know

**Reviewer #4:** Neonatal morbidity and mortality remain high in developing countries. This study analyzed risk factors for preterm birth in developing countries, and measures that can be improved, and the sample size was large. The overall study design is reasonable and the conclusions are reliable. It has important clinical value for reducing neonatal morbidity and mortality in developing countries, and also provides measures for reducing neonatal morbidity and mortality. The authors responded in detail to the comments of the reviewers.

7. PLOS authors have the option to publish the peer review history of their article (what does this mean?). If published, this will include your full peer review and any attached files.

Reviewer #2: No

Reviewer #3: No

Reviewer #4: **Yes: **Guoqiang Cheng

---

## [Author Response · Author response to Decision Letter 1]

31 Dec 2024

Responses to Reviewer 1 Comments:

There were no comments, as this reviewer had already suggested accepting the manuscript at the first revision. 

Responses to Reviewer 2 Comments:

1. Since the elective c-section rate reported is 20% nearly double the rate reported in high-resource countries, and delivery by c-section is associated with LBW, the authors should analyse whether or not elective c-section was an independent risk factor, or at a minimum acknowledge that this is a limitation.

Author response: We agree that the elective caesarean section rate is higher than in the high-resource countries, but it is comparable with the country-specific data. We also agree that elective caesarean section could be associated with LBW. In the study, we did not examine the risk factors for LBW as a whole but examined the risk factors for SGA and prematurity separately. As elective caesarean sections are performed for already determined fetal growth restriction in utero and other indications requiring preterm delivery, elective caesarean section was not included in the regression models evaluating for independent risk factors of SGA and prematurity. We have acknowledged the higher rate of elective caesarean section as a limitation of the study, as suggested by the reviewer (Line 325-330). 

Responses to Reviewer 3 Comments:

2. The bottom line for me is that this study has not added any information that we do not previously know.

Author response: Thank you for your comment. This study is one of the most extensive country-wide studies evaluating the composition and determinants of low birth weight. It describes the relative contribution of prematurity and SGA for LBW in a developing country. It highlights that SGA contributes to the major proportion of LBW in developing countries. Although risk factors for prematurity and LBW are well established, the independent risk factors for SGA determined through large-scale country-wide studies in developing countries are lacking. This study fills that gap by describing the independent risk factors for SGA. We have improved the discussion highlighting this (Line 276-279).

Responses to Reviewer 4 Comments:

3. Neonatal morbidity and mortality remain high in developing countries. This study analysed risk factors for preterm birth in developing countries and measures that can be improved, and the sample size was large. The overall study design is reasonable, and the conclusions are reliable. It has important clinical value for reducing neonatal morbidity and mortality in developing countries and also provides measures for reducing neonatal morbidity and mortality. The authors responded in detail to the comments of the reviewers.

Author response: Thank you very much for the positive comment.

---

## [Decision Letter · Decision Letter 2]

20 Jan 2025

Composition, determinants, and risk factors of low birth weight in Sri Lanka

PONE-D-24-07451R2

Dear Dr. Sachith Mettananda,

We’re pleased to inform you that your manuscript has been judged scientifically suitable for publication and will be formally accepted for publication once it meets all outstanding technical requirements.

Kind regards,

Tamara Sljivancanin Jakovljevic

Academic Editor

PLOS ONE

Reviewers' comments:

Reviewer's Responses to Questions

**Comments to the Author**

1. If the authors have adequately addressed your comments raised in a previous round of review and you feel that this manuscript is now acceptable for publication, you may indicate that here to bypass the “Comments to the Author” section, enter your conflict of interest statement in the “Confidential to Editor” section, and submit your "Accept" recommendation.

Reviewer #2: All comments have been addressed

Reviewer #4: All comments have been addressed

2. Is the manuscript technically sound, and do the data support the conclusions?

Reviewer #2: Yes

Reviewer #4: Yes

3. Has the statistical analysis been performed appropriately and rigorously? 

Reviewer #4: Yes

4. Have the authors made all data underlying the findings in their manuscript fully available?

Reviewer #2: Yes

Reviewer #4: Yes

5. Is the manuscript presented in an intelligible fashion and written in standard English?

Reviewer #2: Yes

Reviewer #4: Yes

6. Review Comments to the Author

Reviewer #4: The author has made a complete revision according to the revised recommendations, which is of great value to how to improve maternal and child health in developing countries, and provides a theoretical basis for the formulation of future improvement measures

---

## [Editor Report · Acceptance letter]

24 Jan 2025

PONE-D-24-07451R2 

PLOS ONE

Dear Dr. Mettananda, 

I'm pleased to inform you that your manuscript has been deemed suitable for publication in PLOS ONE. Congratulations! Your manuscript is now being handed over to our production team.

Kind regards, 

on behalf of

Dr. Tamara Sljivancanin Jakovljevic 

Academic Editor

PLOS ONE
